# Distributed Parallel Gradient Stacking(DPGS):
# Solving Whole Slide Image Stacking Challenge in Multi-Instance Learning

**Boyuan Wu** [1]  **Zefeng Wang** [1]  **Xianwei Lin** [1]  **Jiachun Xu** [2]  **Jikai Yu** [1]  **Shicheng Zhou** [1]  **Hongda Cheng** [1]  **Lianxin Hu** [1]

## Abstract

Whole Slide Image (WSI) analysis is framed as a Multiple Instance Learning (MIL) problem, but existing methods struggle with non-stackable data due to inconsistent instance lengths, which degrades performance and efficiency. We propose a Distributed Parallel Gradient Stacking (DPGS) framework with Deep Model-Gradient Compression (DMGC) to address the problem. DPGS enables lossless MIL data stacking for the first time, while DMGC accelerates distributed training via joint gradient-model compression. Experiments on Camelyon16 and TCGA-Lung datasets demonstrate up to 31× faster training and a maximum 9.3% accuracy improvement compared to baseline. To our knowledge, this is the first work to solve non-stackable data in MIL while improving both speed and accuracy.

## 1. Introduction

Multi-Instance Learning (MIL) has become essential in Whole Slide Image (WSI) analysis for pathological diagnosis, as it can handle bags containing hundreds to thousands of patch instances. WSIs, with ultra-high resolutions (>100,000×100,000 pixels) and sizes up to several GBs, offer rich histological information but introduce significant computational challenges, especially in cancer subtype classification. MIL addresses this by dividing WSIs into patches, selecting informative instances to form variable-length bags, extracting instance-level features, aggregating them into bag-level embeddings, and performing classification. MIL frameworks such as CLAM Lu et al. (2021), which identifies critical regions via interpretable heatmaps, and TransMIL

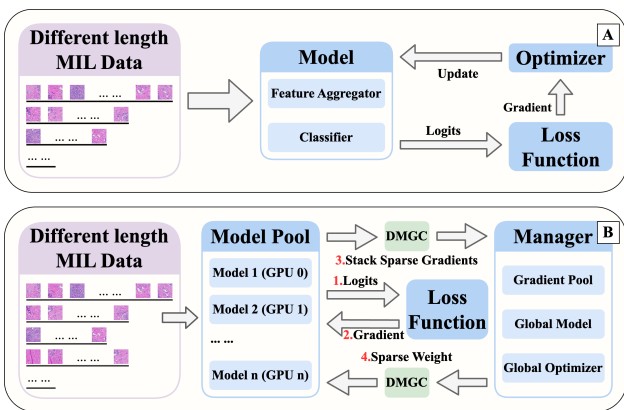

*Figure 1.* **A:** Traditional MIL methods cannot stack data due to the varying lengths of the data. **B:** The proposed framework parallelizes the MIL models and stacks the gradients of these models, thereby simulating data stacking and addressing this issue.

Shao et al. (2021), leveraging Transformers to model instance correlations, exemplify high-accuracy solutions in this domain. However, these methods are unable to address the inherent non-stackability of MIL data. Due to the variation in the number of instances per bag (ranging from hundreds to thousands), conventional approaches are unable to stack bags of different lengths into batches, necessitating sequential processing of individual bags, which brings two critical bottlenecks: (1) Low training efficiency: The inability to utilize GPU parallelism leads to prohibitively long training times for large-scale WSI datasets Wen et al. (2025)Bailly et al. (2022). (2) Unstable gradient estimation: Sequential gradient updates rely on single-bag statistics, introducing bias across non-identically and independently distributed (non-IID) bags and hindering model convergence Li et al. (2019). These two problems severely restrict the performance and scalability of MIL models.

To tackle these challenges, we propose Distributed Parallel Gradient Stacking (DPGS) and Deep Model-Gradient Compression (DMGC). As shown in Figure 1, DPGS assigns variable-length bags to submodels for parallel gradient computation and aggregates gradients to emulate batch stacking without changing the architecture. DMGC compresses gra-

[1]School of Information Engineering, Huzhou University, 313000 Huzhou, China [2]College of Engineering, Carnegie Mellon University, 15213 Pittsburgh, United States of America. Correspondence to: Zefeng Wang <zefeng.wang@zjhu.edu.cn>.

*Proceedings of the $42^{nd}$ International Conference on Machine Learning*, Vancouver, Canada. PMLR 267, 2025. Copyright 2025 by the author(s).

dients and parameters jointly, leveraging sparsity to update only sparse parameters—reducing communication by 99.2% at a 99.99% discard rate while maintaining convergence. On Camelyon16 and TCGA-Lung, DPGS+DMGC improved training speed by up to 31× and accuracy by up to 9.3% over the unstacked MIL baseline.

The main contributions of this study are as follows:

- Distributed Parallel Gradient Stacking for MIL: The first algorithm enabling efficient, lossless MIL data stacking with multi-GPU(or single-GPU) / multi-node training support.

- Deep Model-Gradient Compression: A novel compression strategy jointly optimizing gradient and model parameter sparsity, achieving up to 99.2% communication reduction while maintaining convergence.

- Thoroughly validated on public WSI datasets (Camelyon16, TCGA-Lung) and distributed learning environments, offering an efficient solution for large-scale MIL training.

The paper's organization is as follows: Section II reviews MIL methods, distributed training, and gradient compression; Section III details DPGS and DMGC design with mathematical proofs; Section IV evaluates the framework on multiple WSI datasets across varied configurations; and Section V concludes with future research directions.

## 2. Relate Work

### 2.1. Multi-Instance Learning

MIL provides a framework for analyzing complex data structures containing multiple instances. Existing MIL paradigms typically partition WSIs into localized instances and perform bag-level predictions by aggregating instance-level features. Attention-based MIL methods leverage attention mechanisms for precise feature aggregation, demonstrating strong performance in medical imaging applications Shao et al. (2021); Zhu et al. (2024); Lu et al. (2021); Qu et al. (2024); Tang et al. (2024); Wang et al. (2024b); Ilse et al. (2018). Ilse et al. (2018) pioneered an attention-based MIL architecture that improved both accuracy and interpretability on biomedical datasets. Tang et al. (2024) introduced pseudo-bag construction and dual-layer attention mechanisms to mitigate data scarcity and overfitting, enhancing model generalizability. Graph-based approaches Zheng et al. (2022); Li et al. (2024); Chan et al. (2023); Hou et al. (2022); Zhao et al. (2023) further model instance interactions using graph convolutional networks, achieving state-of-the-art results in cancer subtyping and diagnosis. However, these methods do not address a key MIL limitation: variable bag lengths hinder efficient mini-batch

stacking, reducing computational efficiency, destabilizing gradient estimation, and ultimately degrading convergence speed and model accuracy.

### 2.2. Parallel Computing and Distributed Training

Distributed deep learning employs parallelization to address efficiency and scalability challenges Anil et al. (2018); Dai et al. (2019); Langer et al. (2020). Data parallelism remains the mainstream strategy; Goyal et al. (2017) achieved 90% scaling efficiency across 256 GPUs via linear learning rate scaling. DistBelief Dean et al. (2012) confirmed feasibility for large-scale applications. For models exceeding single-device memory, model parallelism Rasley et al. (2020) partitions parameters across devices. Hybrid methods, such as Megatron-LM Narayanan et al. (2021), combine tensor, pipeline, and data parallelism (PTD-P) to sustain throughput under memory constraints. Decentralized approaches Lin et al. (2017) address privacy concerns via asynchronous updates and the removal of parameter servers Recht et al. (2011).

However, existing frameworks are ill-suited for MIL. Although DistBelief Dean et al. (2012), Megatron-LM Shoeybi et al. (2019), and DeepSpeed Wang et al. (2024a) excel in LLMs and standard tasks, their Transformer-centric designs and batch-based assumptions misalign with MIL. While P-MIL Xu et al. (2017) and MIBP Li et al. (2012) offer tailored solutions, they lack neural MIL compatibility or adaptability to modern models. This incompatibility brings two challenges: (1) architectural mismatch with MIL's instance-level interactions, and (2) inefficient use of resources due to variable-length bag-instance structures that break batch uniformity.

### 2.3. Gradient Compression

Gradient compression techniques reduce communication overhead in distributed training through sparsification Abrahamyan et al. (2021); Chen et al. (2018); Dryden et al. (2016); Wang et al. (2020), quantization, low-rank approximation, and error compensation. Top-$k$ sparsification Aji & Heafield (2017) and Deep Gradient Compression (DGC) Lin et al. (2017) achieve high compression ratios by preserving critical gradients. Quantization methods such as 1-bit SGD Seide et al. (2014) and QSGD Alistarh et al. (2017) balance communication cost with model fidelity. Low-rank approximation Vogels et al. (2019) reduces transmission dimensions via matrix factorization, while error feedback mechanisms Karimireddy et al. (2019) preserve convergence by accumulating untransmitted gradients. Hybrid approaches like DGC Lin et al. (2017) combine sparsification with momentum correction to maintain accuracy under extreme compression.

Despite these advances, gradient compression remains largely unexplored in MIL contexts. Current MIL method-

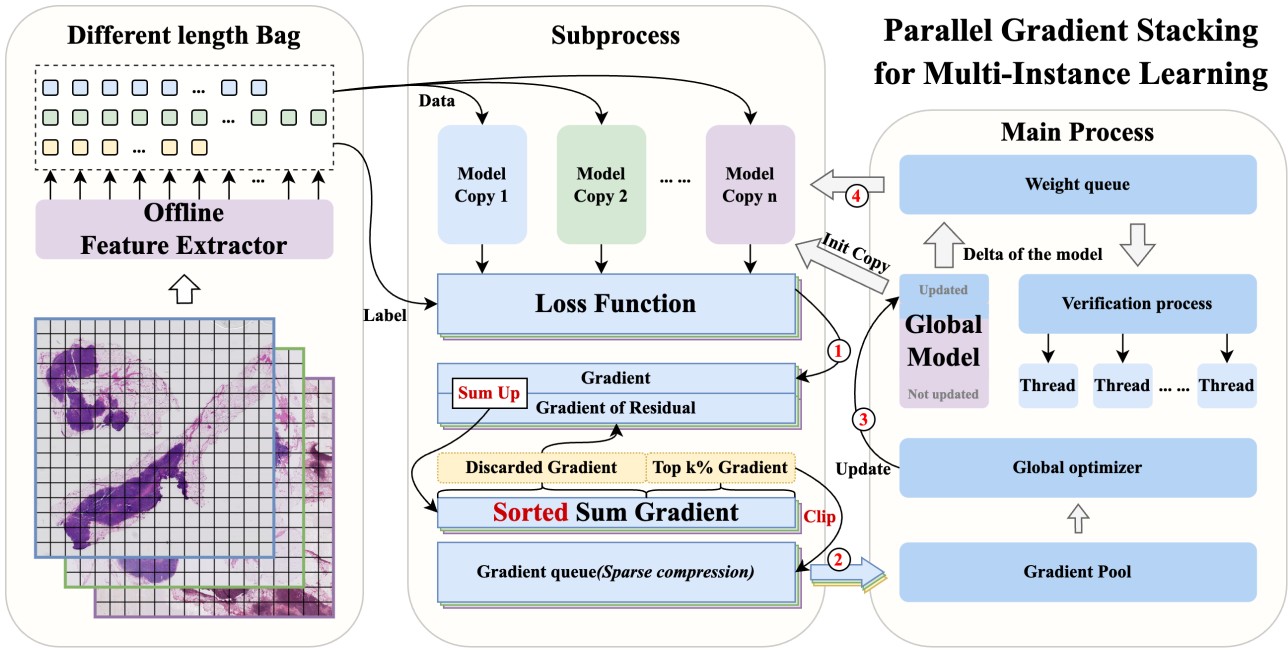

*Figure 2.* The proposed DPGS framework begins by parallelizing models and feeding variable-length data packets for gradient computation. Gradients from parallel models undergo DMGC compression prior to transmission to the master server. The master server aggregates gradients, updates models, applies DMGC compression to weights, and returns them to training servers. Red-numbered annotations in the figure depict the training loop process, with model replication occurring solely during initialization. Multi-threaded validation runs concurrently with the master process.

ologies typically operate in non-distributed settings, overlooking the optimization potential of gradient manipulation in distributed environments.

## 3. Method

This study proposes a gradient-based approach to address MIL sample stacking with inconsistent bag lengths. A distributed gradient aggregation architecture enables parallel processing, improving stacking efficiency and accelerating training. The proposed DMGC reduces bandwidth via dual sparsity (gradients / weights) and improves convergence. These innovations resolve non-stackable data issues and enhance WSI model performance. See Figure 2 and Algorithm 1 for details.

### 3.1. Preliminaries

In MIL, datasets typically consist of multiple bags, with each bag containing several instances. A dataset $\mathcal{D}$ can be represented as $\mathcal{D} = \{(X_j, L_j)\}_{j=1}^N$, where $X_j = \{x_{j,i}\}_{i=1}^{I_j}$ represents the $j$-th bag, which contains $I_j$ instances, and $L_j$ is the label of the bag. Here, $x_{j,i}$ denotes the $i$-th instance of the $j$-th bag. The number of instances $I_j$ may vary across different bags $X$. This inconsistency in bag length makes it challenging to stack the instances of multiple bags into a unified tensor during the training process, thus affecting the

efficiency of parallel computation and model training. For example, consider two bags:

$$X_1 = \{x_{1,1}, x_{1,2}, \ldots, x_{1,I_1}\} \quad (1)$$
$$X_2 = \{x_{2,1}, x_{2,2}, \ldots, x_{2,I_2}\} \quad (2)$$

where $I_1 \neq I_2$. As a result, $X_1$ and $X_2$ cannot be directly stacked into a batch tensor $X$, presenting challenges for parallel computation and accelerating training in MIL models.

### 3.2. Distributed Parallel Gradient Stacking

Given model parameters $\theta^{(t)}$, a sample set $X$ is distributed across $P$ sub-processes $M$ for parallel computation. Each sub-process processes a sub-sample $X_P$, sub-label $L_P$, computes loss $\mathcal{L}_p$ and gradient $\nabla_\theta \mathcal{L}_p$, sends the gradient to the master, and clears its local state without updating. The master using a stacking function $S(\cdot)$, updates the parameters, and broadcasts the updated $\theta^{(t+1)}$. The overall training process $DPGS(\theta^{(t)}, X, P, \eta, M)$ is described as:

$$\theta^{(t+1)} = \theta^{(t)} - \eta \cdot \frac{1}{P} \sum_{p=1}^{P} \nabla_\theta \mathcal{L}_p(M_p(\theta^{(t)}, X_p), L_p) \quad (3)$$

where the gradient stacking function $S(\cdot)$ is defined as:

$$S(\nabla_\theta \mathcal{L}_1, \nabla_\theta \mathcal{L}_2, \ldots, \nabla_\theta \mathcal{L}_P) = \frac{1}{P} \sum_{p=1}^{P} \nabla_\theta \mathcal{L}_p \quad (4)$$

### 3.2.1. EQUIVALENT DERIVATION OF DPGS AND MINI-BATCH

In DPGS, each of the $P$ subprocesses handles a bag $X_p$ and computes the loss $\mathcal{L}_p$ and its gradient $\nabla_\theta \mathcal{L}_p$. If the loss function is an average, the main process averages all subprocess gradients to obtain the global gradient:

$$\nabla_\theta \mathcal{L}_{\text{stacked}} = \frac{1}{P} \sum_{p=1}^{P} \nabla_\theta \mathcal{L}_p \tag{5}$$

In traditional mini-batch training using the mean-based loss function, the $P$ samples $\{X_1, X_2, \ldots, X_P\}$ are stacked into a single mini-batch $B$, and the total loss is computed as:

$$\mathcal{L}_{\text{batch}} = \frac{1}{P} \sum_{p=1}^{P} \mathcal{L}_p \tag{6}$$

Then, the mini-batch gradient is calculated:

$$\nabla_\theta \mathcal{L}_{\text{batch}} = \nabla_\theta \left( \frac{1}{P} \sum_{p=1}^{P} \mathcal{L}_p \right) = \frac{1}{P} \sum_{p=1}^{P} \nabla_\theta \mathcal{L}_p \tag{7}$$

By comparison, we find:

$$\nabla_\theta \mathcal{L}_{\text{batch}} = \nabla_\theta \mathcal{L}_{\text{stacked}} \tag{8}$$

Thus, these two methods are equivalent in model training. After obtaining the gradients, we update the model weights:

$$\theta^{(t+1)} = \theta^{(t)} - \eta \cdot \left( \frac{1}{P} \sum_{p=1}^{P} \nabla_\theta^t \mathcal{L}_p \right) \tag{9}$$

Here, we use SGD for the mathematical equivalence derivation, but in practice, any optimizer can be used to update the model.

### 3.3. Deep Model-Gradient Compression

Distributed training incurs high bandwidth usage due to gradient and weight transmission Dean et al. (2012). DGC Lin et al. (2017) addresses this by sending only large-magnitude gradients and accumulating residuals, achieving dual sparsity. We propose DMGC, an improved version of DGC that transmits weight deltas instead of full weights (Algorithm 1), further reducing bandwidth without sacrificing training stability.

---

**Algorithm 1** DPGS Algorithm with DMGC

**Input:**
$X$ : Input data samples
$Y$ : Ground truth labels for the training data
$P$ : Number of subprocesses for parallel computation
$\eta$ : Learning rate
$M$ : Model function used for training
$k$ : Top-K ratio for gradient compression
$thr_{clip}$ : Gradient clipping threshold
Initialize parameters: $\theta^{(0)}, \theta^{Comp}, res_p = 0$
**Training Service:**
**while** not converged **do**
  $\theta^{DMGC} \leftarrow Decompress(\theta^{Comp})$
  $\theta^{(t)} \leftarrow \theta^{(t-1)} + \theta^{DMGC}$
  $res_p \leftarrow 0$
  **for** each $p = 1 \ldots P$ **do**
    $X_p \leftarrow$ Subset of $X$
    $G_p \leftarrow \nabla_\theta \mathcal{L}(M(\theta^{(t)}, X_p), Y_p)$
    $G_p^{comb} \leftarrow G_p + res_p$
    **for** each $i \in len(G_p^{comb})$ **do**
      $thr_p \leftarrow k\%$ of $|G_p^{comb}|$
      $mask \leftarrow |G_p^{comb}[i]| > thr_p$
      $G_p^{mask} \leftarrow G_p^{comb}[i] \times mask$
      $res_p[i] \leftarrow G_p^{comb}[i] - G_p^{mask}$
    **end for**
    $G_p^{DMGC} \leftarrow Clip(G_p^{mask}, thr_{clip})$
    $G^{Comp} \leftarrow Compress(G_p^{DMGC})$
    Send $G^{Comp}$ to master
    $G_p \leftarrow 0$
  **end for**
  Wait for synchronization with Master Service
**end while**
**Master Service:**
**while** not converged **do**
  **for** each $p = 1 \ldots P$ **do**
    Receive $G^{Comp}$ from Training Service
    $G_p^{DMGC} \leftarrow Decompress(G^{Comp})$
  **end for**
  $G_{global} \leftarrow \eta \cdot \frac{1}{P} \sum_{p=1}^{P} G_p^{DMGC}$
  $\theta^{(t+1)} \leftarrow \theta^{(t)} - \eta \cdot G_{global}$
  $\theta^{DMGC} \leftarrow \theta^{(t+1)} - \theta^{(t)}$
  $\theta^{Comp} \leftarrow Compress(\theta^{DMGC})$
  Send $\theta^{Comp}$ to all Training Service
  Check convergence condition
**end while**

---

### 3.3.1. VIRTUAL BATCH IN DMGC

DMGC and related locally accumulated gradient compression methods (e.g., DGC) transmit only gradients exceeding a preset threshold, while accumulating sub-threshold updates locally until they surpass it, drastically reducing communication bandwidth. Through multi-round accumulation,

this process is mathematically equivalent to using larger batch sizes. Within the DPGS framework, the learning rate can still follow linear scaling rules despite gradient accumulation. The following derivation clarifies this behavior under DPGS.

Consider the update process for a specific component $\theta^{(i)}$ of the model parameters $\theta$. Assume there are $P$ subprocesses, each processing a batch $X_p$ and computing its corresponding loss $\mathcal{L}_p$ and gradient $\nabla_\theta \mathcal{L}_p$. After introducing DMGC within the DPGS framework, only gradients that exceed the threshold are transmitted, while other gradients are accumulated locally in the subprocesses. After $T$ local gradient accumulations, a global update is performed, and the mathematical expression for this is:

$$\theta_{t+T}^{(i)} = \theta_t^{(i)} - \eta T \cdot \frac{1}{PbT} \sum_{p=1}^{P} \left( \sum_{\tau=0}^{T-1} \nabla^{(i)} \mathcal{L}_p^{(t+\tau)} \right) \quad (10)$$

Here, $\eta$ represents the learning rate, $P$ is the number of subprocesses, $b$ is the batch size each subprocess handles (in MIL, $b = 1$), and $T$ is the gradient accumulation interval. This equation indicates that within a gradient update interval of $T$, each subprocess accumulates gradients from $T$ iterations, which are then applied to a single global update. Originally, the effective batch size per iteration is $P \cdot b$, but after $T$ accumulations, it is equivalent to expanding the batch size to $P \cdot b \cdot T$. Therefore, each global update implicitly uses a larger virtual batch size of $P \cdot b \cdot T$.

In large-batch training, learning rate scaling is typically required for stability. However, as shown above, the learning rate $\eta \times T$ and the scaled batch size $P \cdot b \cdot T$ effectively cancel out. Thus, methods based on local gradient accumulation (e.g., DMGC) inherently follow the linear scaling rule Goyal et al. (2017), requiring no significant adjustment to maintain stability and convergence.

### 3.4. Time Complexity Analysis

We analyze the computational efficiency of the DPGS framework by comparing time complexities with conventional sequential training, noting that reduced time complexity does not directly imply faster end-to-end training since gradient quality and optimizer dynamics also influence convergence rates.

Let $N$ be the number of bags, with $T_{\text{forward}}$, $T_{\text{backward}}$, and $T_{\text{update}}$ denoting the per-bag times for forward, backward propagation, and parameter updates. Communication overhead is $T_{\text{comm}} \propto M \cdot (1/C_{\text{ratio}})$, where $M$ is the number of parameters, and $C_{\text{ratio}}$ is the gradient compression ratio.

Sequential training's total time is:

$$T_{\text{serial}} = N \cdot (T_{\text{forward}} + T_{\text{backward}} + T_{\text{update}}), \quad (11)$$

due to the inefficiency of processing variable-length bags.

DPGS enhances efficiency via gradient stacking and distributed computation. Each epoch processes $P$ bags in $\lceil N/P \rceil$ rounds. Each round includes: 1. Parallel forward-backward passes synchronized by the slowest gradient computation, costing $\max(T_{\text{forward}} + T_{\text{backward}})$. 2. Sparse gradient aggregation, costing $T_{\text{comm}} \cdot (1/C_{\text{ratio}}) + T_{\text{update}}$. 3. Global parameter updates, fixed at $T_{\text{update}}$.

The parallel time complexity is:

$$T_{\text{parallel}} = \frac{N}{P} \left( \max(T_{\text{fwd}} + T_{\text{bwd}}) + T_{\text{c}}/C_{\text{r}} + T_{\text{u}} \right). \quad (12)$$

The theoretical speedup ratio is approximately:

$$\frac{P}{1+\alpha}, \quad (13)$$

where

$$\alpha = \frac{T_{\text{comm}} \cdot M \cdot (1/C_{\text{ratio}}) + T_{\text{update}}}{\max(T_{\text{forward}}, T_{\text{backward}})}, \quad (14)$$

with near-linear scaling when $\alpha \ll 1$.

This analysis solely addresses per-iteration time reduction. Actual training duration depends on convergence speed. While increasing effective batch size via gradient accumulation (e.g., $P \cdot T$ in DPGS) may reduce noise and accelerate convergence, excessive noise suppression risks suboptimal training, causing empirical rates to deviate from linear scaling. For experimental results, please refer to 4.2.2

## 4. Experiments

### 4.1. Data

*Table 1.* Original data distribution of the datasets employed. Notably, the Camelyon 16 test set utilizes an officially predefined split, comprising 38.92% tumor class and 61.71% normal tissue class. In contrast, the TCGA-Lung dataset was randomly partitioned with an 8:2 training-to-test ratio. IFL:Instance Feature Length

| CLASSIFICATION | TYPE I | TYPE II | IFL |
|---|---|---|---|
| C16 MULTISCALE | 160 40.11% | 239 59.89% | 512 |
| C16 IMAGENET | 160 40.11% | 239 59.89% | 256 |
| TCGA-LUNG MULTISCALE | 534 51.05% | 512 48.95% | 1024 |
| TCGA-LUNG UIN2 | 534 51.05% | 512 48.95% | 1536 |

Experiments on two public WSI datasets, using varied feature extraction strategies, effectively yielded four datasets for analysis: Camelyon16 (C16) with single-scale ResNet50 (ImageNet-pretrained) and multiscale features, and TCGA-Lung with multiscale features obtained from DS-MIL Li et al. (2021) and single-scale features extracted by UIN2.

*Table 2.* Experiments compare MIL methods' accuracy/convergence time with/without DPGS on C16-Multiscale, C16-ImageNet, and TCGA-Lung Multiscale, demonstrating DPGS's dual improvement in accuracy and convergence speed. Table annotations: " — " indicates model non-convergence (accuracy equals original data distribution, no convergence time recorded)* , while "B" specifies batch size at achieved performance levels.

| DATA | | C16 MULTISCALE | | TCGA-LUNG MULTISCALE | | C16 IMAGENET PRETRAIN | |
|---|---|---|---|---|---|---|---|
| FRAMEWORK | METHOD | ACCURACY | TIME (S) | ACCURACY | TIME (S) | ACCURACY | TIME (S) |
| CLASSIC | ABMIL[ILSE ET AL. (2018)] | $91.47\%_{\pm 0.84}$ | $405.5^{(B=16)}_{\pm 11.1}$ | $92.38\%_{\pm 0.23}$ | $636.3^{(B=32)}_{\pm 12.7}$ | $75.05\%_{\pm 1.82}$ | $1177.2^{(B=16)}_{\pm 33.5}$ |
| | MeanMIL | $61.71\%^*$ | - | $\mathbf{92.23\%}_{\pm 0.49}$ | $883.1^{(B=32)}_{\pm 67.7}$ | $61.71\%^*$ | - |
| | TransMIL[SHAO ET AL. (2021)] | $93.80\%_{\pm 0.32}$ | $304.2^{(B=16)}_{\pm 6.1}$ | $92.39\%_{\pm 0.22}$ | $397.5^{(B=32)}_{\pm 17.9}$ | $80.48\%_{\pm 2.7}$ | $1163.1^{(B=16)}_{\pm 46.6}$ |
| | CLAM-MB[LU ET AL. (2021)] | $93.02\%_{\pm 0.96}$ | $499.3^{(B=16)}_{\pm 19.9}$ | $92.60\%_{\pm 0.39}$ | $503.2^{(B=32)}_{\pm 10.1}$ | $78.91\%_{\pm 3.6}$ | $1152.1^{(B=16)}_{\pm 53.8}$ |
| | ACMIL[ZHANG ET AL. (2024)] | $94.40\%_{\pm 0.64}$ | $953.4^{(B=16)}_{\pm 88.7}$ | $93.13\%_{\pm 0.84}$ | $711.2^{(B=32)}_{\pm 61.23}$ | $81.31\%_{\pm 0.41}$ | $2735.3^{(B=16)}_{\pm 46.32}$ |
| | RRTMIL[TANG ET AL. (2024)] | $94.57\%_{\pm 0.85}$ | $2462.1^{(B=16)}_{\pm 15.5}$ | $93.26\%_{\pm 0.42}$ | $1923.8^{(B=32)}_{\pm 41.4}$ | $82.53\%_{\pm 0.31}$ | $3099.1^{(B=16)}_{\pm 21.5}$ |
| PADDING | ABMIL | $92.23\%_{\pm 0.58}$ | $1512.8^{(B=16)}_{\pm 40.2}$ | $91.43\%_{\pm 0.26}$ | $1465.0^{(B=32)}_{\pm 99.3}$ | $82.48\%_{\pm 2.79}$ | $9040.5^{(B=16)}_{\pm 280.8}$ |
| | MeanMIL | $61.71\%^*$ | - | $90.00\%_{\pm 0.33}$ | $1722.1^{(B=32)}_{\pm 64.4}$ | $61.71\%^*$ | - |
| | TransMIL | $94.04\%_{\pm 0.34}$ | $2912.8^{(B=16)}_{\pm 108.3}$ | $93.82\%_{\pm 0.60}$ | $1054.1^{(B=32)}_{\pm 91.1}$ | $84.38\%_{\pm 2.22}$ | $9754.0^{(B=16)}_{\pm 295.1}$ |
| | CLAM-MB | $93.80\%_{\pm 0.42}$ | $2737.1^{(B=16)}_{\pm 64.7}$ | $92.00\%_{\pm 0.37}$ | $1063.2^{(B=32)}_{\pm 21.3}$ | $83.28\%_{\pm 1.30}$ | $9700.1^{(B=16)}_{\pm 194.0}$ |
| | ACMIL | $92.63\%_{\pm 0.61}$ | $798.6^{(B=16)}_{\pm 35.79}$ | $93.33\%_{\pm 0.87}$ | $2077.8^{(B=32)}_{\pm 99.1}$ | $84.77\%_{\pm 0.36}$ | $9552.4^{(B=16)}_{\pm 103.7}$ |
| | RRTMIL | $93.02\%_{\pm 0.96}$ | $1299.3^{(B=16)}_{\pm 19.9}$ | $93.62\%_{\pm 0.72}$ | $2872.1^{(B=32)}_{\pm 85.2}$ | $84.90\%_{\pm 0.21}$ | $10023.1^{(B=16)}_{\pm 192.1}$ |
| SAMPLING | ABMIL | $84.58\%_{\pm 1.56}$ | $3811.9^{(B=4)}_{\pm 23.1}$ | $92.45\%_{\pm 0.59}$ | $3564.7^{(B=8)}_{\pm 18.6}$ | $73.84\%_{\pm 1.50}$ | $22054.2^{(B=4)}_{\pm 101.0}$ |
| | MeanMIL | $61.71\%^*$ | - | $87.46\%_{\pm 3.22}$ | $4016.4^{(B=8)}_{\pm 36.3}$ | $61.71\%^*$ | - |
| | TransMIL | $89.93\%_{\pm 5.13}$ | $3721.6^{(B=4)}_{\pm 84.3}$ | $92.33\%_{\pm 3.79}$ | $4316.3^{(B=8)}_{\pm 97.2}$ | $79.89\%_{\pm 4.23}$ | $30281.9^{(B=4)}_{\pm 293.8}$ |
| | CLAM-MB | $86.24\%_{\pm 3.92}$ | $3012.7^{(B=4)}_{\pm 21.1}$ | $92.24\%_{\pm 3.22}$ | $4416.2^{(B=8)}_{\pm 89.1}$ | $75.15\%_{\pm 4.15}$ | $28231.3^{(B=4)}_{\pm 221.1}$ |
| | ACMIL | $91.86\%_{\pm 3.28}$ | $1088.8^{(B=4)}_{\pm 81.3}$ | $93.34\%_{\pm 3.92}$ | $6461.2^{(B=8)}_{\pm 98.1}$ | $84.62\%_{\pm 3.91}$ | $20112.2^{(B=4)}_{\pm 231.5}$ |
| | RRTMIL | $91.74\%_{\pm 7.04}$ | $6050.6^{(B=4)}_{\pm 29.2}$ | $93.66\%_{\pm 3.01}$ | $6901.2^{(B=8)}_{\pm 100.3}$ | $84.79\%_{\pm 4.06}$ | $32211.2^{(B=4)}_{\pm 425.1}$ |
| DPGS+DMGC | ABMIL | $\mathbf{94.04\%}_{\pm 0.14}$ | $\mathbf{29.3}^{(B=16)}_{\pm 2.6}$ | $\mathbf{93.90\%}_{\pm 0.13}$ | $\mathbf{20.0}^{(B=32)}_{\pm 2.4}$ | $\mathbf{84.38\%}_{\pm 1.04}$ | $\mathbf{92.4}^{(B=8)}_{\pm 11.2}$ |
| | MeanMIL | $71.33\%_{\pm 0.02}$ | $104.1^{(B=16)}_{\pm 9.1}$ | $92.23\%_{\pm 0.02}$ | $14.1^{(B=32)}_{\pm 2.3}$ | $65.63\%_{\pm 0.90}$ | $122.2^{(B=16)}_{\pm 19.1}$ |
| | TransMIL | $\mathbf{95.23\%}_{\pm 0.22}$ | $\mathbf{115.2}^{(B=16)}_{\pm 12.3}$ | $\mathbf{94.70\%}_{\pm 0.41}$ | $\mathbf{48.6}^{(B=32)}_{\pm 6.9}$ | $\mathbf{85.97\%}_{\pm 1.49}$ | $\mathbf{158.1}^{(B=8)}_{\pm 15.2}$ |
| | CLAM-MB | $\mathbf{93.82\%}_{\pm 0.64}$ | $\mathbf{28.1}^{(B=16)}_{\pm 2.5}$ | $\mathbf{93.23\%}_{\pm 0.19}$ | $\mathbf{40.1}^{(B=16)}_{\pm 7.6}$ | $\mathbf{83.59\%}_{\pm 1.23}$ | $\mathbf{109.2}^{(B=8)}_{\pm 9.2}$ |
| | ACMIL | $\mathbf{95.87\%}_{\pm 0.41}$ | $\mathbf{49.9}^{(B=16)}_{\pm 12.8}$ | $\mathbf{94.79\%}_{\pm 0.49}$ | $\mathbf{67.3}^{(B=32)}_{\pm 11.2}$ | $\mathbf{87.71\%}_{\pm 0.52}$ | $\mathbf{61.2}^{(B=16)}_{\pm 17.2}$ |
| | RRTMIL | $\mathbf{95.97\%}_{\pm 0.79}$ | $\mathbf{615.7}^{(B=16)}_{\pm 16.5}$ | $\mathbf{94.90\%}_{\pm 0.38}$ | $\mathbf{570.1}^{(B=32)}_{\pm 22.1}$ | $\mathbf{87.99\%}_{\pm 0.26}$ | $\mathbf{710.2}^{(B=16)}_{\pm 38.1}$ |

Multiscale features were derived from multiple magnifications (e.g., $20\times$, $5\times$) and concatenated into a feature pyramid. This approach preserves tissue details across scales while optimizing feature usage via local attention, enhancing both classification and lesion localization. TCGA-Lung UIN2 refers to features extracted by the UIN2 feature extractor on the TCGA dataset Chen et al. (2024). As shown in Table 1, the scale, distribution, and feature diversity of these datasets provide a robust benchmark for evaluating model robustness.

### 4.2. Results of DPGS

#### 4.2.1. DPGS VS PADDING VS SAMPLING VS CLASSIC

This study evaluates the DPGS framework and DMGC through controlled experiments using three datasets from Section 4.1. Metrics include classification accuracy and convergence time, defined per Jahani-Nasab & Bijarchi (2024) as the point when total loss remains below the 95th percentile across trials. Our experiments compare three baselines: 1. Classic 2. Padding: Applies zero-padding to shorter bags for uniform length. 3. Sampling: Randomly selects a fixed number of instances per bag.

Experimental Setup: 4 nodes, 10% DMGC retention, $4\times$NVIDIA V100 GPUs (32GB), 1000 Mbps bandwidth (non-parallel baselines use a single GPU).

As shown in Table 2, DPGS boosts ABMIL accuracy on TCGA-LUNG from 92.38% to 93.90% and speeds up convergence by $31.8\times$ (636.3s to 20.0s). On Camelyon16-Multiscale, MeanMIL improves by 9.62% (61.71% to 71.33%). All baselines show accuracy and convergence gains, highlighting DPGS's generalizability. Table 3 confirms further performance improvements with the modern feature extractor UNI2 Chen et al. (2024).Further evidence of the generalisability of our approach.

*Table 3.* Performance of TCGA-LUNG Features Extracted by UNI2 on ABMIL

| Dataset | TCGA-UNI2 | |
|---|---|---|
| Framework | ACC | Time |
| Classic | $95.30\%_{\pm 0.71\%}$ | $895.96^{(B=1)}_{\pm 25.76}$ |
| Padding | $96.10\%_{\pm 0.65\%}$ | $2767.9^{(B=32)}_{\pm 54.42}$ |
| Sampling | $95.12\%_{\pm 0.11\%}$ | $233.75^{(B=32)}_{\pm 17.25}$ |
| DPGS+DMGC | $\mathbf{96.90\%}_{\pm 0.31\%}$ | $\mathbf{32.10}^{(B=32)}_{\pm 4.31}$ |

Further analysis reveals Bag Padding's limitations: while achieving a 2.37%–7.43% accuracy boost on Camelyon16-ImageNet-Pretrain, it suffers a 3.2–8.4$\times$ slower convergence compared to other baselines. We attribute this to: (1) excessive zero-padding lowering information density

and complicating convergence, and (2) data dimensional expansion increasing computation and memory costs. Table 2 confirms DPGS's dual advantage over Bag Padding in both accuracy and convergence, validating its optimized training dynamics and improved generalization.

For Sampling, although this approach accelerates convergence, it reduces classification accuracy and increases variance (see supplementary table). This may compromise data integrity—mathematically distinct from standard minibatch sampling—as it violates positive bag completeness and risks omitting critical instances, especially in larger batches Shapcott et al. (2019). In contrast, our method achieves a better trade-off between efficiency and accuracy.

Notably, variations in bag length (57k/105k for C16 Multiscale/ImageNet) significantly impact processing speed, explaining the latency observed with Bag Pooling. Dataset size also affects the optimal batch size (TCGA=32 vs. C16=16), highlighting the importance of tuning hyperparameters based on scale. Differences in method performance become less pronounced with simpler features but more evident as feature complexity increases.

### 4.2.2. THE NODE EXPANSION EFFICIENCY OF DPGS

This study provides a theoretical and empirical analysis of scalability in distributed training. As shown in Figure 3, with fixed batch size (B = 8), DMGC keep rates (10%, 1%, 0.1%), and 1000 Mbps bandwidth on the C16-Multiscale dataset, ABMIL training time decreases nearly linearly as node count increases (K = 1–8), but shows significant non-linear reduction beyond K = 4 (at 1% keep rate). Aligned

*Table 4.* A comparison of ABMIL performance in single- vs. multi-machine training shows that the method supports single-GPU operation via multi-process parallelism (virtual nodes), maximizing resource use without requiring multi-GPU setups. Intra-device communication improves efficiency, leveraging memory bandwidth that exceeds network limits.

| Dataset | C16-Multiscale | |
| --- | --- | --- |
| Configure | 4 Nodes on 1 GPU | 4 Nodes on 4 GPUs |
| Acc | 93.71% ± 0.11% | 93.83% ± 0.09% |
| Time | 133.32 ± 17.47 | 142.95 ± 11.73 |

with Equation 12, we posit that this phenomenon originates from fundamental scalability constraints of distributed systems: when the node scale surpasses $K > 4$ (keep rate = 1%), the proportion of cross-node gradient synchronization time ($T_{\text{comm}}$) in the total training duration progressively increases. At this critical juncture, the diminishing returns from reduced computation time per iteration are offset by the rising communication overhead, pushing overall training efficiency into a plateau phase.

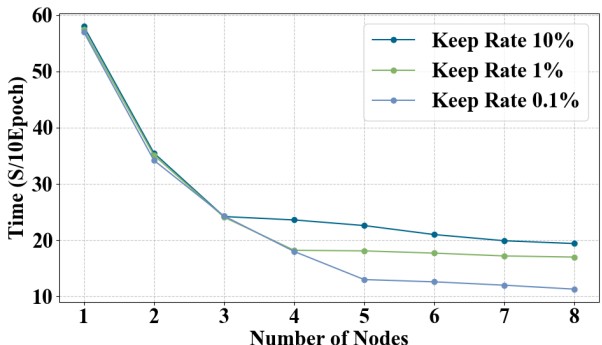

*Figure 3.* Synchronous parallel algorithms face scalability plateaus due to communication and synchronization overheads. While expanding nodes reduces training costs, this gain is offset by increased communication. Higher gradient compression shifts the plateau inflection point to larger scales. In addition, as per Equation 12, larger datasets prolong computational benefits from node scaling under fixed communication costs, delaying the plateau and raising the bottleneck inflection point.

As demonstrated in Table 4, it is important to recognize that the proposed methodology is not limited to multi-node distributed systems; it can also be implemented on single-node computing devices. By using multi-process virtualization to simulate nodes, DPGS can operate in a distributed manner on a single device. Furthermore, the improved communication efficiency within a single machine's memory can lead to higher computational efficiency.

### 4.2.3. BATCH SIZE GAIN FOR DPGS

This study further investigates the regulatory mechanism of batch size on MIL model performance. Under traditional non-parallel frameworks constrained by hardware limitations that preclude batch stacking, we systematically analyze the dynamic relationship between batch scale and model efficacy within the DPGS framework by varying gradient stacking magnitudes under identical experimental configurations.

Figure 4 presents the results across the three datasets discussed in Section 4.1. In the TCGA-LUNG dataset, all comparative models exhibit monotonically increasing accuracy as the stacking quantity increases from 8 to 32. However, performance degrades when the batch size exceeds a critical threshold ($N = 32$). Notably, model sensitivity to batch size varies significantly. When the batch size increases from 4 to 32 in the Camelyon16-Multiscale dataset, accuracy and convergence time trends diverge across models. Moreover, optimal batch sizes vary substantially between datasets. Comparative analysis reveals a notable distinction between TCGA-LUNG (simple-structured data) and Camelyon16-ImageNet-Pretrain (complex, lower-quality data): models

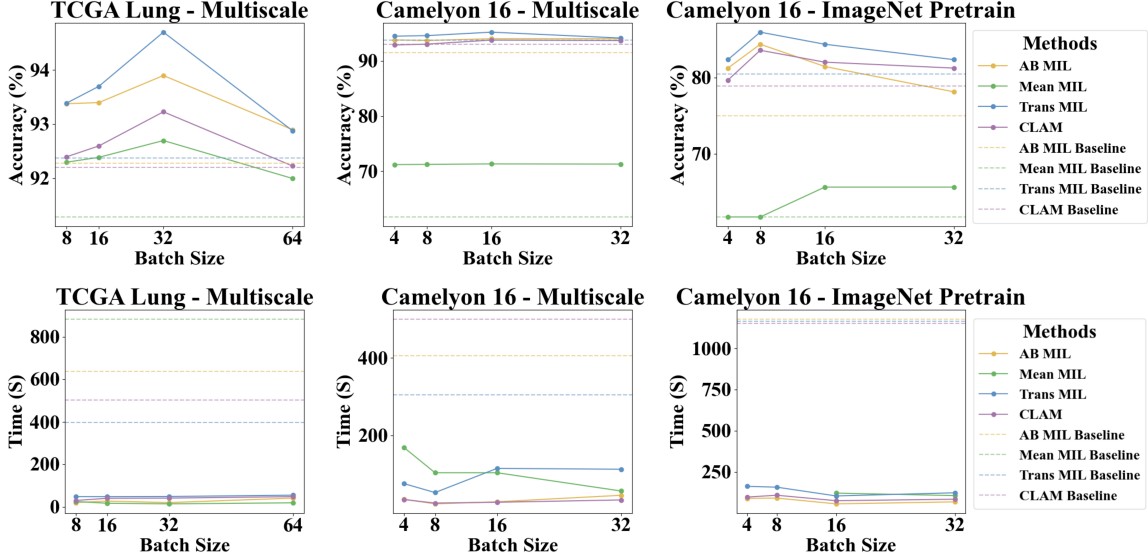

*Figure 4.* The dashed line signifies the baseline of accuracy and time when this framework is not employed. The incomplete line indicates that the model did not converge on this dataset, consequently, a convergence time is not available. The corresponding accuracy is represented by the high value in the test data distribution.

yield significantly greater accuracy gains (4.68%–9.33% vs. 0.4%–1.3%) with batch size scaling in the complex dataset.

This discrepancy is attributable to two factors: (1) Complex datasets offer more room for accuracy improvement due to lower baseline performance; (2) Stacking-driven error estimation reduces gradient noise more effectively, improving the consistency of parameter updates, especially in noise-intensive gradients associated with complex data patterns.

### 4.3. Results of DMGC

#### 4.3.1. ABLATION STUDY FOR DMGC

This study systematically validates the core advantages of DMGC through controlled ablation experiments. Under fixed network bandwidth (20 Mbps) and a computational node scale of $N = 4$ on the C16-Multiscale dataset, five independent experimental replicates were conducted, and mean values along with standard deviations were calculated. As shown in Table 5, when using the combined DPGS+DMGC strategy, the ABMIL and TransMIL models achieve 73.9% and 76.2% reductions in convergence time, respectively, compared to the DPGS baseline methods, while maintaining identical classification accuracy.

#### 4.3.2. BANDWIDTH AFFECTS DMGC

To investigate algorithmic robustness under network bandwidth constraints, we establish controlled network environments using switch-based rate limiting and conduct multi-bandwidth experiments on ABMIL models with the C16-Multiscale dataset. Figure 5 shows that under various low-

*Table 5.* An ablation analysis was conducted to evaluate the accuracy and convergence time of ABMIL and TransMIL integrated with DPGS under a 20 Mbps network bandwidth. The experimental results demonstrate that the DPGS framework combined with DMGC achieves the fastest convergence speed while maintaining competitive accuracy performance.

| Model | Method | Accuracy | Time (s) |
|---|---|---|---|
| ABMIL | DPGS+None | $93.84\%_{\pm 0.08}$ | $544_{\pm 37}$ |
| | DPGS+DGC[Lin et al. (2017)] | $93.83\%_{\pm 0.12}$ | $249_{\pm 15}$ |
| | DPGS+DMGC | $93.83\%_{\pm 0.09}$ | $\mathbf{142}_{\pm 11}$ |
| TransMIL | DPGS+None | $95.23\%_{\pm 0.06}$ | $1311_{\pm 90}$ |
| | DPGS+DGC | $95.20\%_{\pm 0.08}$ | $652_{\pm 55}$ |
| | DPGS+DMGC | $95.23\%_{\pm 0.04}$ | $\mathbf{313}_{\pm 19}$ |

bandwidth conditions, DMGC achieves superior convergence speed compared to both DGC and non-compression methods, with average acceleration factors of $1.73\times$ over DGC and $3.08\times$ over uncompressed approaches. These results confirm that DMGC maintains stable model performance under stringent bandwidth limitations while significantly improving communication efficiency compared to DGC.

#### 4.3.3. DISCARD RATE IMPACTS ON DMGC.

This study analyzes gradient discard rates in DMGC through controlled experiments. As Table 6 showed Under 10Mbps bandwidth constraints, DMGC maintains stable convergence even at 99.99% discard rates, demonstrating excep-

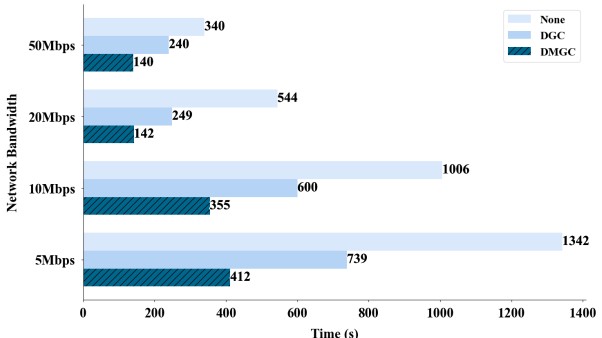

*Figure 5.* The convergence times of the ABMIL model with different compression methods on the C16-Multiscale dataset under varying bandwidths. It should be noted that the accuracy in this experiment remained nearly unchanged, with a variance of less than 0.03%. It can be observed that as the bandwidth decreased, the advantage of this method over DGC and the non-compression approach became increasingly pronounced.

tional robustness. A non-monotonic convergence pattern emerges: optimal speed at 99% discard rate vs. suboptimal performance at 30%, stemming from the balance between 1. Low discard rates preserve gradient tensor semantic integrity but significantly increase network transmission latency; 2. High discard rates reduce communication overhead at the cost of increased gradient sparsification.

Further analysis demonstrates DMGC's dual compression (gradient sparsification + weight sparsification) significantly improves communication efficiency. Experiments show 117.5-fold parameter compression ratio versus baseline. However, diminishing marginal returns in compression efficiency are observed in low discard rate regimes. This phenomenon occurs due to heterogeneous sparse weights across distributed nodes. Only parameters that exceed retention thresholds are updated, which reduces weight compressibility and leads to a decline in marginal utility.

## 5. Conclusion

This study introduces DPGS, a distributed MIL framework that integrates gradient stacking with dual compression (DMGC), achieving the following:

- 1. Lossless batch processing of variable-length data via distributed gradients.

- 2. A 99.2% bandwidth reduction through joint gradient-weight compression, as shown in Table 6.

- 3. Up to $31\times$ faster training and 9.3% accuracy improvements over baselines on the Camelyon16 and TCGA-Lung datasets.

While DPGS optimizes the efficiency of synchronous dis-

*Table 6.* Impact of DMGC Dropout Rates on ABMIL Performance(C16-Multiscale,10Mbps). Comm: Communication size(Byte). CR: Compression ratio(original size / compressed size)

| Keep Rate | Time (s) | Acc. (%) | Comm. ($\times$1e3) | CR |
|---|---|---|---|---|
| 0.01% | $5256_{\pm 226}$ | $93.8_{\pm 0.2}$ | $8.3_{\pm 0.4}$ | 117.5 |
| 0.1% | $1380_{\pm 181}$ | $93.8_{\pm 0.1}$ | $10.7_{\pm 0.2}$ | 91.6 |
| 1% | $\mathbf{355}_{\pm 60}$ | $93.8_{\pm 0.2}$ | $42.9_{\pm 5.4}$ | 22.8 |
| 10% | $409_{\pm 35}$ | $94.0_{\pm 0.1}$ | $90.4_{\pm 6.5}$ | 10.8 |
| 20% | $470_{\pm 33}$ | $94.0_{\pm 0.0}$ | $244.1_{\pm 7.3}$ | 4.4 |
| 30% | $531_{\pm 34}$ | $94.0_{\pm 0.0}$ | $384.5_{\pm 7.9}$ | 2.5 |
| 40% | $564_{\pm 33}$ | $93.8_{\pm 0.2}$ | $440.0_{\pm 7.5}$ | 2.2 |
| 50% | $621_{\pm 30}$ | $93.8_{\pm 0.0}$ | $697.2_{\pm 9.0}$ | 1.4 |
| 60% | $445_{\pm 26}$ | $93.8_{\pm 0.0}$ | $733.1_{\pm 8.6}$ | 1.3 |
| 70% | $\underline{390}_{\pm 26}$ | $94.0_{\pm 0.0}$ | $884.9_{\pm 9.6}$ | 1.1 |
| 80% | $441_{\pm 29}$ | $93.8_{\pm 0.1}$ | $923.5_{\pm 9.6}$ | 1.1 |
| 90% | $724_{\pm 37}$ | $93.8_{\pm 0.2}$ | $937.6_{\pm 10.8}$ | 1.0 |
| 100% | $731_{\pm 34}$ | $93.8_{\pm 0.1}$ | $979.7_{\pm 17.5}$ | 1.0 |

tributed training, the synchronization overhead still limits GPU utilization. Future work will explore asynchronous distributed computing to mitigate synchronization latency and further accelerate large-scale MIL workflows. Combining adaptive task scheduling with asynchronous updates has the potential to improve the scalability of ultra-large histopathology datasets, paving the way for distributed MIL at petabyte scale.

## Impact Statement

This paper presents work whose goal is to advance the field of Machine Learning. There are many potential societal consequences of our work, none which we feel must be specifically highlighted here.

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
