# OpenReview forum: "Distributed Parallel Gradient Stacking(DPGS): Solving Whole Slide Image Stacking Challenge in Multi-Instance Learning"
_ICML.cc/2025/Conference — ICML 2025 poster_

### Official Review · Reviewer_psN8 · 2025-03-13

**Overall Recommendation:** 3

**Summary:**

This paper proposes a Distributed Parallel Gradient Stacking (DPGS) framework and a Deep Model Gradient Compression (DMGC) technique to address the non-stackable data issues caused by variable bag lengths in whole slide image (WSI) multiple instance learning (MIL). The core innovations include: 1) Enabling lossless batch processing through distributed sub-model parallel computation and gradient aggregation; 2) Jointly compressing gradients/parameters and leveraging gradient sparsity to update only sparse parameters, reducing communication overhead by 99.2% while maintaining convergence; 3) Experimental results demonstrate 31× acceleration in training speed and 9.3% improvement in accuracy on Camelyon16 and TCGA-Lung datasets.

**Claims And Evidence:**

Most of the claims made in the manuscript are supported by clear and convincing evidence. The shortcoming is that the motivation of this paper is not strongly substantiated. For instance, the article claims, "There are two critical bottlenecks: (1) Slow training speed: The inability to utilize GPU parallelism leads to prohibitively long training times for large-scale WSI datasets; (2) Unstable gradient estimation: Sequential gradient updates rely on single-bag statistics, introducing bias across non-identically and independently distributed (non-IID) bags and hindering model convergence." However, the second point lacks support from both references and experimental evidence.

**Essential References Not Discussed:**

To the best of my knowledge, the key contribution of the proposed Distributed Parallel Gradient Stacking (DPGS) framework in this paper is unique. It introduces the DPGS framework combined with Deep Model Gradient Compression (DMGC) technology to address the issue of non-stackable data caused by varying bag lengths in Whole Slide Image (WSI) multiple instance learning. The related work is comprehensively reviewed, but the motivation presented in the introduction may require further support from additional literature or experimental evidence.

**Experimental Designs Or Analyses:**

Yes, I have checked the validity of experimental designs and analyses from Section 4.1 to Section 4.3. The experimental design and analysis conducted for the proposed method in the article are methodologically sound and empirically valid.

**Methods And Evaluation Criteria:**

Yes, the proposed method(s) and/or evaluation criteria (e.g., benchmark datasets) are appropriately justified for the current problem or application.

**Other Comments Or Suggestions:**

Since I am not familiar with work in the field of distributed computing or training in the general machine learning area, I do not have any other comments or suggestions.

**Other Strengths And Weaknesses:**

Strengths:
1.The paper introduces a novel approach (DPGS) to address the critical challenge of non-stackable data in MIL by leveraging distributed gradient stacking instead of traditional bag padding, enabling lossless MIL batch processing while improving both speed and accuracy
2.The proposed method achieves up to 31× faster training and 9.3% accuracy gains on widely used medical datasets (Camelyon16, TCGA-Lung). This holds a certain significance for the development of foundational models that require large-scale pre-training in the field of computational pathology.

Weaknesses:
As I am not familiar with work in the field of distributed computing or training in the general machine learning area, I conservatively point out the following shortcomings of the work:
1. The introduction of the paper mentions a key bottleneck of unstable gradient estimation faced by traditional MIL model training. To address this issue, the paper designed experiments to compare the convergence times of different methods. However, it is not clear how “convergence” is defined, which may not be sufficient to support the claim that the instability of gradient estimation and the difficulty of convergence in previous methods have been resolved.
2. The explanation as to why existing distributed training frameworks (such as Megatron-LM, DeepSpeed) or gradient compression methods (such as DGC) cannot be directly applied to MIL models is not clear.

**Questions For Authors:**

1.Although bags of different lengths cannot be directly concatenated into a batch tensor, to my knowledge, the built-in collate_fn function in PyTorch’s DataLoader can load multiple bags within a single batch, which means that batch training is possible on a single GPU. What advantages does the proposed DPGS offer over this training approach?
2.How does DMGC ensure that gradients critical for early-stage convergence (e.g., large-magnitude or directionally consistent gradients) are retained during sparsification?

**Relation To Broader Scientific Literature:**

The Distributed Parallel Gradient Stacking (DPGS) framework proposed in this paper represents the first work to achieve simultaneous improvements in both speed and accuracy while addressing the issue of non-stackable data in Multiple Instance Learning (MIL). This holds significant importance for the current development of large-scale pre-trained pathology foundation models. The training method introduced in this paper has the potential to enhance the performance of existing Slide-level pre-trained foundation models, such as those referenced in [1][2][3].

[1] A whole-slide foundation model for digital pathology from real-world data (Nature 2024)
[2] A pathology foundation model for cancer diagnosis and prognosis prediction (Nature 2024)
[3] SlideChat: A Large Vision-Language Assistant for Whole-Slide Pathology Image Understanding (CVPR 2025)

**Theoretical Claims:**

Yes, I have examined the principles of the method proposed in the article, primarily the Deep Model-Gradient Compression outlined in Section 3.3. After reviewing the references cited in this section (Dean et al., Lin et al.), I find the principles underlying the proposed method to be sound and without issue.

---

> ### Author Rebuttal · Authors · 2025-03-31
>
> Q1: Most of the claims made in the manuscript are supported by clear and convincing evidence. The shortcoming is that the motivation of this paper is not strongly substantiated.
>
> A1:
> We sincerely appreciate the reviewer's valuable feedback.
> Regarding training efficiency, we clarify that "excessively long" was intended to highlight the relative efficiency difference between existing methods and our approach under the same conditions. We have added reference [1] to support this.
> For the second point, we agree that a clearer theoretical link between non-IID data and convergence difficulties is needed. We’ve cited [2], which proves: (1) non-IID data causes gradient bias, delaying or preventing convergence, and (2) appropriate gradient aggregation strategies (e.g., mean normalization) mitigate this effect. Our gradient aggregation mechanism is based on these insights.
>
> Q2:
>     However, it is not clear how “convergence” is defined, which may not be sufficient to support the claim that the instability of gradient estimation and the difficulty of convergence in previous methods have been resolved.
>
> A2:
> We sincerely appreciate the reviewer's valuable feedback.
> We define convergence using the threshold-based criterion from [3], where the model is considered converged when the total loss remains consistently below a fixed threshold. This method aligns with standard practices and ensures objective stability assessment. We derived the threshold by analyzing the 95th percentile of the loss distribution across multiple trials, ensuring robustness.
>
> Q3：
>     The explanation as to why existing distributed training frameworks (such as Megatron-LM, DeepSpeed) or gradient compression methods (such as DGC) cannot be directly applied to MIL models is not clear.
>
> A3:
> We sincerely appreciate the reviewer's valuable feedback.
>     The existing distributed frameworks (e.g., Megatron-LM/DeepSpeed) indeed excel in traditional LLM scenarios but face two fundamental challenges when applied to MIL:
> (1) Architectural Compatibility: Current frameworks are primarily optimized for structured architectures like Transformers (e.g., tensor parallelism with intra-layer partitioning, as in Megatron-LM). However, MIL’s multi-instance interaction mechanism, due to its distinct design philosophy and structure, requires different parallelization strategies, making these frameworks suboptimal for MIL.
> (2) Data Characteristics: MIL’s hierarchical "bag-instance" data (variable-length, non-directly stackable) violates the conventional distributed training assumption of "batch homogeneity." This leads to inefficient data sharding and memory wastage when using existing frameworks. Our proposed method is specifically tailored to MIL’s unique computational paradigms and data topology.
> Thus, in the MIL domain, existing distributed training frameworks cannot be efficiently applied.
>
> Q4:
>     Although bags of different lengths cannot be directly concatenated into a batch tensor, to my knowledge, the built-in collate_fn function in PyTorch’s DataLoader can load multiple bags within a single batch, which means that batch training is possible on a single GPU. What advantages does the proposed DPGS offer over this training approach?
>
> A4:
> We sincerely appreciate the reviewer's valuable feedback.
> Although PyTorch’s collate_fn provides flexibility, preprocessing steps like Bag Pooling are still needed to form batch tensors, potentially causing information loss and additional overhead. DPGS eliminates these steps, as shown in Table 2, where it significantly outperforms pooling-based methods in efficiency.
>
> Q5：How does DMGC ensure that gradients critical for early-stage convergence (e.g., large-magnitude or directionally consistent gradients) are retained during sparsification?
>
> A5:
> We sincerely appreciate the reviewer's valuable feedback.
> DMGC preserves essential gradients through:
> (1) Magnitude-based Gradient Filtering: We retain the top k% largest gradients, which dominate optimization (Table 4 shows that even retaining just 0.01% allows convergence).
> (2) Momentum Accumulation: Historical gradients stabilize training by suppressing noise and fluctuations. These mechanisms collectively preserve key gradients while minimizing ineffective updates.
>
> Thank you for your valuable feedback. If you have further questions, feel free to ask. We would really appreciate it if you find our revisions satisfactory and adjust your evaluation.
>
> [1]Wen, Jiangping, Jinyu Wen, and Emei Fang. "MsaMIL-Net: An End-to-End Multi-Scale Aware Multiple Instance Learning Network for Efficient Whole Slide Image Classification." arXiv preprint arXiv:2503.08581 (2025).
>
> [2]Li, Xiang, et al. "On the convergence of fedavg on non-iid data." arXiv preprint arXiv:1907.02189 (2019).
>
> [3]Jahani-Nasab, Mahyar, and Mohamad Ali Bijarchi. "Enhancing convergence speed with feature enforcing physics-informed neural networks using boundary conditions as prior knowledge." Scientific Reports 14.1 (2024): 23836.

---

> > ### Comment · Reviewer_psN8 · 2025-04-03
> >
> > The rebuttal addresses my concerns and I'll retain the score.

---

### Official Review · Reviewer_7pdf · 2025-03-13

**Overall Recommendation:** 3

**Summary:**

This paper proposes a framework called DPGS combined with DMGC to address the non-stacked data problem in MIL, particularly in WSI analysis. DPGS addresses the inefficiencies due to variable-length instance bags that prevent effective batch stacking by parallelizing MIL models and stacking gradients instead of raw data, thereby achieving significant training speedups. DMGC further enhances performance by compressing both gradients and model weights. Experimental results on Camelyon16 and TCGA-Lung datasets demonstrate up to 31× faster training and 9.3% accuracy improvement over baseline models.

**Claims And Evidence:**

The paper claims to enable lossless data stacking in MIL, accelerate training, and improve model accuracy. However, the experimental data appear to contain discrepancies. For instance, inconsistencies exist between the experimental data presented in Table 2 and the descriptions within the main text. Notably, the convergence time of the Bag padding method in Table 2 significantly exceeds that of the Classic method (non-parallel methods restricted to a single GPU), which appears to be anomalous. Furthermore, the precise methodology for computing the convergence time evaluation metric is not adequately elucidated.

**Essential References Not Discussed:**

The paper provides a comprehensive overview of related works. Given the paper's emphasis on distributed training and gradient compression, incorporating recent studies on asynchronous distributed MIL and federated MIL frameworks would furnish a more comprehensive perspective of the field.

**Experimental Designs Or Analyses:**

The experimental design and analysis are generally robust. However, the validation of model accuracy could benefit from the inclusion of stronger baselines. Furthermore, it is imperative to ensure the veracity of the experimental data, eliminating any typographical errors within the tables, as these significantly undermine the credibility of the reported results.

**Methods And Evaluation Criteria:**

The methodology proposed in this manuscript offers the potential to accelerate the training of giga-pixel Whole Slide Image (WSI) analysis, thereby possessing significant research implications

**Other Comments Or Suggestions:**

Clarify the trade-offs between gradient stacking and asynchronous updates.

**Other Strengths And Weaknesses:**

Strengths:
- The paper proposes DPGS to stack giga-pixel Whole Slide Images (WSIs) with inconsistent instance lengths, thereby accelerating training.
- Experimental results demonstrate that the proposed method significantly enhances both training speed and accuracy.

Weaknesses:
- The novelty is somewhat incremental, as the method leverages established gradient accumulation, distributed training, and gradient compression techniques.
- The computation methodology for metrics such as convergence time remains undefined.
- Numerous data errors significantly compromise the credibility of the experimental results.

**Questions For Authors:**

-  Why is the Time data for the MEANMIL method unavailable in Table 2?
- Why is the DPGS+DMGC method not compared under identical "B" conditions?

**Relation To Broader Scientific Literature:**

The manuscript applies widely adopted techniques, such as gradient accumulation, distributed training, and gradient compression, to the task of giga-pixel WSI analysis. The connections to prior works are well-articulated.  A discussion on how DPGS compares to recent federated MIL approaches would strengthen the literature review.

**Theoretical Claims:**

N/A. The demonstration of equivalent derivations between DPGS and traditional mini-batch training is straightforward.

---

> ### Author Rebuttal · Authors · 2025-04-01
>
> Q1:Table 2 shows Bag Padding's unexpectedly long convergence time versus Classic. The convergence time calculation method also requires clarification.
>
> A1:Thank you for reviewing.Bag Padding's longer convergence time (Table 2) stems from large packet length spans (C16 MULTISCALE: 223-57,318; C16 IMAGENET: 487-105,775), requiring excessive padding and computation time that outweighs batch benefits. Larger batches mitigate but don't eliminate this penalty.
>
> Q2:Comparing DPGS with recent federated MIL approaches and including asynchronous distributed MIL studies would enhance the literature review's comprehensiveness.
>
> A2:Thank you for reviewing.We acknowledge the need to compare with async distributed/federated MIL methods. Following your guidance, we systematically searched Web of Science using:"Federated Learning" AND "multi-instance learning""asynchronous distributed" AND "multi-instance learning"Both queries returned null results, further confirming our study's novelty.
>
> Q3:The method's novelty is limited, building on existing gradient accumulation, distributed training, and compression techniques.
>
> A3:Thank you for reviewing.We appreciate the reviewer's rigorous evaluation. While acknowledging prior work (gradient accumulation/distributed training), our core contribution addresses MIL's fundamental bottleneck - non-stackable data - which has constrained recent NN-based MIL studies to inefficient batch_size=1 training[2][3].Our three-level innovation:Problem: First lossless MIL data stacking enabling batch_size scalability (orders-of-magnitude efficiency gain)Method: DMGC extends DGC via gradient sparsity for compressed weight distribution (lower communication overhead)Framework: First adaptation of distributed concepts to MIL's variable-length sequences (bridging general/MIL-specific methods)
> We fully agree these innovations build upon existing technologies, much like how Transformer builds upon self-attention mechanisms. However, proposing systematic solutions to domain-specific critical problems constitutes significant innovation in itself.
>
> Q4:While the experimental design is robust, adding stronger baselines would improve accuracy validation.
>
> A4:Thank you for reviewing.To address your suggestions and improve rigor, we added comparisons with two recent MIL models (RRTMIL, CVPR 2024[2]; ACMIL, ECCV 2024[3]). Preliminary results (below) show our framework retains significant training efficiency advantages over these models (ACMIL1.47%ACC and 19.1XTime improve/RRTMIL 1.4%ACC and 3.99XTime improve). (Full table:https://drive.google.com/file/d/1VODIlXC0Qd1wXPao16yJFCReRAYj3MBV/view?usp=sharing)Ongoing experiments will be fully reported in the revision.
>
> Q5:DPGS+DMGC: Why no same-B comparison?MEANMIL:Why missing time data in Table 2?
>
> A5:Thank you for reviewing.Our tiered B-value design (C16: 1/4/8/16/32; TCGA-LUNG: 1/8/16/32/64) reflects real-world possible batch size selection. Classic only supports B=1 (no stacking),therefore, no comparison can be made; we'll correct Bag Padding labels and provide (https://drive.google.com/file/d/1_DUaRWWV7AcXjODUHF9RfFp_GsJFThd-/view?usp=sharing).For MeanMIL's missing timing:Unstable convergence prevented standard timing calculation [1].Non-convergent cases used class priors (marked *) as conservative estimates.Convergence time shown as "-" with explanatory notes.
>
> Q6:Clarify the trade-offs between gradient stacking and asynchronous updates.
>
> A6:Thank you for reviewing.Our design employs synchronous distributed training rather than asynchronous updates due to two key factors:
> Asynchronous updates cause stale gradients that led to significant training instability in our preliminary experiments.
> While synchronous updates require coordination overhead, our DMGC method minimizes communication costs while preserving stability through gradient stacking.
>
> We greatly appreciate your feedback. Should you have any additional questions, please don't hesitate to ask. We'd be most grateful if you find our revisions satisfactory and consider adjusting your evaluation.
>
> [1]Jahani-Nasab, Mahyar, and Mohamad Ali Bijarchi. "Enhancing convergence speed with feature enforcing physics-informed neural networks using boundary conditions as prior knowledge." Scientific Reports 14.1 (2024): 23836.
> [2]Tang, Wenhao, et al. "Feature re-embedding: Towards foundation model-level performance in computational pathology." Proceedings of the IEEE/CVF Conference on Computer Vision and Pattern Recognition. 2024.
> [3]Zhang, Yunlong, et al. "Attention-challenging multiple instance learning for whole slide image classification." European Conference on Computer Vision. Cham: Springer Nature Switzerland, 2024.

---

> > ### Comment · Reviewer_7pdf · 2025-04-07
> >
> > Thank you for the responses, which have largely addressed my concerns. After careful consideration of all comments and the corresponding responses, I have decided to raise my score to weak accept.

---

### Official Review · Reviewer_g8fv · 2025-03-13

**Overall Recommendation:** 3

**Summary:**

This paper introduces Distributed Parallel Gradient Stacking (DPGS), a framework designed to address the challenge of non-stackable data in Multiple Instance Learning (MIL) for Whole Slide Image (WSI) analysis. The authors propose two key components: (1) DPGS, which enables parallel processing of variable-length MIL bags by distributing them across multiple GPUs and aggregating their gradients, and (2) Deep Model-Gradient Compression (DMGC), which reduces communication overhead during distributed training through joint compression of gradients and model parameters. Experiments on Camelyon16 and TCGA-Lung datasets demonstrate significant improvements in both training speed (up to 31× faster) and classification accuracy (up to 9.3% increase) compared to baseline methods.

**Claims And Evidence:**

The paper makes several claims that are supported by experimental evidence but lack critical context:
1.	Problem significance: The authors frame non-stackable MIL data as a critical bottleneck for training efficiency, claiming that sequential processing leads to "prohibitively long training times." However, from my experience, many current MIL methods can be trained on a single high-memory GPU (e.g., RTX3090 with 24GB memory) in reasonable timeframes (half a day) for the datasets used in this paper.
2.	Acceleration and accuracy improvement: The paper claims up to 31× faster training and 9.3% accuracy improvement. While the experimental results in Table 2 support these numbers, the comparison is against sequential processing methods without consideration of simpler alternatives like uniform sampling approaches that could enable standard batch training.
3.	Mathematical equivalence of DPGS to mini-batch training: The mathematical derivations showing DPGS is equivalent to mini-batch SGD are sound, but this equivalence raises questions about why standard mini-batch approaches with fixed-size bags (through patch sampling) couldn't achieve similar results with less complexity.
4.	Communication efficiency of DMGC: The ablation studies support the claim that DMGC achieves significant communication reduction. However, this actually leads to another major question that existing works can be trained with a single GPU, yet the proposed method requires multi-GPU for training. In clinical settings, researcher may not have access to multi-GPU setups.

**Essential References Not Discussed:**

The Related Work section in this paper is comprehensive, yet some minor discussion is needed:
1.	The paper doesn't adequately discuss simpler alternatives to handling variable-length bags, such as uniform sampling strategies.
2.	The MIL approaches used as baselines do not represent the current state-of-the-art.

**Experimental Designs Or Analyses:**

Please address the questions in the Methods and Evaluation Criteria section.
Additional questions:
1.	Limited analysis of computational requirements: There is insufficient analysis of how results might change with different hardware configurations, particularly more accessible setups with fewer or less powerful GPUs.
2.	Impact of foundation models: In the field of computational pathology, multiple foundation models have been proposed yet I see no use here in this paper. From the community’s experience, using foundation models as feature extractors and a simple MIL model (such as ABMIL), the performance on these two datasets could easily goes up to near 100%. I believe it would be better to take these into consideration.

**Methods And Evaluation Criteria:**

The idea of this work is timely and interesting. And the method design seems reasonable. However, the evaluation has several significant limitations:
1.	Unclear feature extraction: The "multi-scale" features referenced for both datasets are not adequately defined, making it impossible to understand their relationship to current feature extraction approaches in computational pathology.
2.	Outdated baselines: The paper considers older MIL methods (ABMIL, MeanMIL, TransMIL, CLAM-MB) while omitting including more recent and advanced MIL approaches that represent the current state-of-the-art in computational pathology and pathology foundation models for feature extraction.
3.	Missing comparisons to simpler alternatives: The paper did not evaluate simpler baselines such as uniform sampling that could potentially address the variable-length bag issue with much less complexity and computational overhead.
4.	Limited dataset analysis: While the paper uses established datasets (Camelyon16 and TCGA-Lung), it could further benefit from adequately discussing the dataset characteristics or how they might influence the observed results.

**Other Comments Or Suggestions:**

1.	Typos should be carefully filtered: for example, in the first paragraph in Introduction, “followed by classification.MIL framework”.
2.	Meanwhile, please check the indent in the paper. For example, in the first paragraph of Introduction, “exemplify high-accuracy solutions in this domain.However,”.
3.	Could you put Figure 1 at the top of Page 1?

**Other Strengths And Weaknesses:**

Strengths:
1.	The mathematical derivations showing equivalence to mini-batch training are sound.
2.	The gradient compression approach (DMGC) offers an interesting extension to existing gradient compression techniques.
3.	The ablation studies provide useful insights into the factors affecting performance within their framework.
Weaknesses:
1.	High resource requirements: Although the speed-up performance is impressive, the method requires multiple GPUs and high-bandwidth connections, significantly limiting its practical applicability in many research and clinical settings.
2.	Missing comparisons to simpler alternatives: The paper doesn't evaluate simple alternatives such as uniform sampling that could potentially achieve similar results with much less complexity.
3.	Outdated baselines: The paper relies on comparisons with older MIL methods rather than current state-of-the-art approaches.
4.	Unclear feature extraction: The "multi-scale" features referenced throughout the paper are not adequately defined.
5.	Limited relevance given foundation models: The paper doesn't acknowledge or compare against foundation models that have demonstrated near-perfect performance on the same datasets.

**Questions For Authors:**

1.	Could you provide a clear definition of the "multi-scale" feature extraction process used in your experiments, including architectures and implementation details?
2.	Have you compared your approach with simpler methods like uniform sampling from each WSI to create fixed-length bags that could be trained with standard batch processing?
3.	Could you explain your experimental setting and the reason why you did not consider current foundation model-based approaches (UNI [1], CONCH [2], PLIP [3], etc.) that have demonstrated state-of-the-art performance on the same datasets?
4.	What is the minimum hardware configuration required to achieve meaningful benefits from your approach compared to single-GPU training?
5.	Why did you choose to compare against older MIL methods rather than more recent approaches that might represent stronger baselines?
I will consider raising the overall recommendation score if these questions are solved in the rebuttal phase.
[1] Chen, Richard J., et al. "Towards a general-purpose foundation model for computational pathology." Nature Medicine 30.3 (2024): 850-862.
[2] Lu, Ming Y., et al. "A visual-language foundation model for computational pathology." Nature Medicine 30.3 (2024): 863-874.
[3] Huang, Zhi, et al. "A visual–language foundation model for pathology image analysis using medical twitter." Nature medicine 29.9 (2023): 2307-2316.

**Relation To Broader Scientific Literature:**

The proposed method contributes to the entire community of computational pathology, facilitating efficient and effective frameworks for whole slide image analysis. The speed-up in terms of training time and batch training are interesting and considered for the modern MILs for the first time.

**Theoretical Claims:**

The theoretical foundations of the paper are generally sound.

---

> ### Author Rebuttal · Authors · 2025-04-01
>
> Q1:The paper calls non-stackable MIL data a bottleneck citing 'prohibitively long training times',yet current methods train adequately on single GPUs within half-day for these datasets
>
> A1:Thank you for reviewing.Clarifications:This is an inexact error "long training times":Our method shows superior efficiency under identical conditions. While some MIL methods use 24GB GPUs, repeated runs (tuning/ablation) require 10-100× more time,making efficiency critical.Current medical datasets (thousand-scale) remain smaller than natural image sets (e.g., ImageNet).As [1] shows,larger datasets improve generalization,suggesting medical data will grow.Our framework accelerates training through multi-node/single-GPU parallelism and enables federated learning for multi-center collaboration,getting ready for the future
>
> Q2:The reported performance improvements lack comparisons with simpler alternatives like uniform sampling that could potentially achieve comparable results more efficiently.
>
> A2:Thank you for reviewing.For benchmarks, we added uniform sampling tests (currently ABMIL-only due to time constraints).Results show it speeds convergence but reduces accuracy and increases variance(see table:https://drive.google.com/file/d/1sjgW2z6Y9pbgwI9v6TwaJBZcwBd4PC4O/view?usp=sharing),compromising data integrity—mathematically distinct from standard minibatching by breaking positive bag completeness and potentially missing key instances,with larger batches worsening these effects [2] .Our method maintains better efficiency-accuracy tradeoffs.
>
> Q3: While DMGC enhances communication efficiency, its multi-GPU dependency and unproven performance on typical clinical hardware (especially low-GPU setups) pose practical concerns.
>
> A3:Thank you for reviewing.To clarify:Our method supports single-GPU operation through multi-process parallelization (virtual nodes) to maximize resource utilization.This optimizes usage without multi-GPU setups. Intra-device communication enhances efficiency as memory bandwidth exceeds network limits. We compared ABMIL performance using 4 nodes on single 4060 8G vs. multi-machine V100 32G configurations(https://drive.google.com/file/d/1biUVuB6COd79X3NADdsiwm9va2p1Mpyd/view?usp=sharing), with detailed implementation analysis to be included in revision.
>
> Q4:Unclear feature extraction:The"multi-scale"features referenced for both datasets are not adequately defined,obscuring their relation to current computational pathology feature extraction methods.
>
> A4:Thank you for reviewing.Multi-scale features were obtained from DS-MIL's repository (https://github.com/binli123/dsmil-wsi/issues/49), extracted at multiple magnifications (e.g.20×,5×) and concatenated into a feature pyramid.This maintains tissue details across scales while optimizing feature use via local attention, enhancing both classification and lesion localization.
>
> Q5:Outdated baselines:The paper uses older MIL methods but omits recent advanced MIL approaches representing current state-of-the-art in computational pathology and pathology foundation models for feature extraction.
>
> A5:Thank you for reviewing.Key clarifications:Our focus is a general MIL framework addressing non-stackable data bottlenecks, not a new model, hence using classical models (ABMIL,TransMIL) for comparison.New tests compare with recent models (RRT-MIL,ACMIL), showing training efficiency gains (ACMIL1.47%ACC and 19.1XTime improve/RRTMIL 1.4%ACC and 3.99XTime improve)(Full table::https://drive.google.com/file/d/1VODIlXC0Qd1wXPao16yJFCReRAYj3MBV/view?usp=sharing).Due to time constraints,the full experiments will be provided in the revised version.
>
> Q6:Limited dataset analysis:While using established datasets, the paper should better discuss their characteristics and potential impact on results.
>
> A6: Key findings:Bag length variation (Δ57k/105k for C16 Multiscale/ImageNet) affects speed, explaining Bag Pooling's lag.Dataset size dictates optimal batch (TCGA=32 vs C16=16), showing scale's hyperparameter impact.Method differences fade with easier features (TCGA) but intensify with hard ones (C16 ImageNet).
>
> Q7:The study's omission of current foundation models (e.g., UNI, CONCH, PLIP) that achieve near-perfect performance on these datasets with simple MIL architectures requires justification.
>
> A7:Thank you for reviewing.Our core contribution addresses MIL's data stacking limitations through framework innovation (not feature extraction), justifying initial internal comparisons.Following your advice, we've added comparisons(https://drive.google.com/file/d/1Xs2YqWd7SCp4Te8PaqnUbjXEzdf3p6Vd/view?usp=sharing);Due to time,the full experiments will be provided in the revised version.
>
> [1]Bailly,et al. "Effects of dataset size and interactions on the prediction performance of logistic regression and deep learning models." Computer Methods and Programs in Biomedicine 213 (2022): 106504.
> [2]Shapcott M,et al.Deep Learning With Sampling in Colon Cancer Histology.Front Bioeng Biotechnol.2019;7:52.

---

> > ### Comment · Reviewer_g8fv · 2025-04-07
> >
> > Thanks for the detailed response. The updated results in the rebuttal seem promising, with 20x speeding up and performance gain at the same time. I see that the authors have tried their best to answer the questions and most of them are solved. Although the writing quality of this paper requires further polishing, I decide to raise the score to weak accept.

---

### Official Review · Reviewer_M5Bd · 2025-03-14

**Overall Recommendation:** 3

**Summary:**

The manuscript introduces a framework designed to overcome the challenges posed by variable-length bags in Multiple Instance Learning (MIL) for whole slide image analysis. The framework was called Distributed Parallel Gradient Stacking (DPGS), distributes the gradient computation across multiple processes and stacking gradients. This method simulates mini-batch training without the need for data padding. The authors also propose Deep Model-Gradient Compression (DMGC), which jointly compresses both gradients and model weights to reduce communication overhead significantly. Experiments on the Camelyon16 and TCGA-Lung datasets indicate improvements in training speed and accuracy over standard approaches.

**Claims And Evidence:**

The submission does support its claims with both theoretical derivations and empirical evidence. The authors provide theoretical Justification that gradient stacking is mathematically equivalent to traditional mini-batch training. The empirical results on Camelyon16 and TCGA-Lung datasets show improvements in convergence time and prediction accuracy.

**Essential References Not Discussed:**

No.

**Experimental Designs Or Analyses:**

The combination of DPGS and DMGC addresses the challenge of handling variable-length bags in MIL. Experiments conducted on Camelyon16 and TCGA-Lung provide a measure of the proposed method’s effectiveness in real-world scenarios. The evaluation criteria include both prediction accuracy and convergence time, which assesses not just the model’s performance but also its efficiency.

**Methods And Evaluation Criteria:**

The combination of DPGS and DMGC addresses the challenge of handling variable-length bags in MIL. Experiments conducted on Camelyon16 and TCGA-Lung provide a measure of the proposed method’s effectiveness in real-world scenarios. The evaluation criteria include both prediction accuracy and convergence time, which assesses not just the model’s performance but also its efficiency.

**Other Comments Or Suggestions:**

No.

**Other Strengths And Weaknesses:**

The paper enables efficient parallel processing despite variable instance lengths in MIL. The combinations of DPGS and DMGC improve training efficiency and model accuracy.

The manuscript provides a derivation demonstrating the equivalence of gradient stacking to traditional mini-batch training. The time complexity analysis strengthen the theoretical foundation of the approach.

The results, presented in tables and figures, effectively demonstrate both convergence speed improvements and accuracy gains.

The integration between DPGS and DMGC could be better articulated. It is unclear how the compression interacts with the distributed gradient stacking.

Additional details regarding hyperparameter choices (e.g., selection of compression thresholds, batch sizes in different settings) would further strengthen reproducibility.

**Questions For Authors:**

1. Could you clarify the integration between DPGS and DMGC, specifically how the compression interacts with the distributed gradient stacking process?

2. Could you provide additional details regarding your hyperparameter choices—such as the selection of compression thresholds and the batch sizes used in different settings?

**Relation To Broader Scientific Literature:**

The paper bridges gaps between MIL-specific challenges and the broader advances in distributed deep learning and gradient compression.

**Theoretical Claims:**

The authors derive that aggregating gradients computed on separate bags (via DPGS) is mathematically equivalent to computing the gradient on a mini-batch formed by stacking the data. The authors also present a theoretical time complexity analysis that compares the parallel training time with sequential training. The derivation provides a reasonable estimate of speedup and highlights potential scalability.

---

> ### Author Rebuttal · Authors · 2025-03-31
>
> Q1：Could you clarify the integration between DPGS and DMGC, specifically how the compression interacts with the distributed gradient stacking process?
>
> A1：
>     We sincerely appreciate the reviewer's valuable feedback. We apologize for the lack of clarity in describing the relationship between the components in our original manuscript. Regarding your question, we provide the following clarification:
>     The DMGC module is responsible for the compression and transmission of gradients within the gradient-based DPGS framework. Specifically, during distributed training, after worker nodes have completed their local gradient computations, the gradients are first processed by the DMGC module for compression before being transmitted to the main server. Consequently, the main server receives not the full gradients but rather the sparsified gradients compressed by DMGC.
>     On the main server side, DPGS aggregates these compressed sparse gradients to form new global gradients. It is noteworthy that due to the high compression ratio of DMGC, the aggregated gradients remain sparse (the extent of sparsity is contingent on the keep rate). This inherent sparsity enables selective model updates, i.e. only a subset of weights require updating rather than full parameter updates. The system leverages this property to transmit only the sparse weight updates back to the worker nodes, thereby significantly reducing communication overhead (this represents one of DMGC's key innovations compared to DGC[1]).
>
> We sincerely appreciate your valuable feedback. If you have any further questions about the paper, please do not hesitate to ask us.
>
> Q2：Could you provide additional details regarding your hyperparameter choices—such as the selection of compression thresholds and the batch sizes used in different settings?
>
> A2：
>     We sincerely appreciate the reviewer's valuable feedback.  Regarding your question, we provide the following clarification:
>     Regarding the experiments presented in Table 2 and Figure 4, all DPGS+DMGC tests were conducted with 4 worker nodes and 1 master process, using a fixed gradient compression ratio of 10% (retaining 10% of gradients; results with different compression ratios can be found in Table 4). All frameworks (except the Classic framework which cannot support batch stacking) were evaluated with multiple batch sizes, employing a stepwise B-value configuration adapted to each dataset's scale (C16: B=1/4/8/16/32; TCGA-LUNG: B=1/8/16/32/64). The complete DPGS+DMGC results across different B-values are presented in Figure 4.
>     Within the DPGS+DMGC framework shown in Table 2 and Figure 4, the master process maintains a fixed gradient buffer size of 4, while worker nodes dynamically adjust their gradient accumulation steps according to the batch size (e.g., when batch size is 16, both the gradient buffer and accumulation steps are set to 4, making the effective gradient update equivalent to batch size 16). All experiments used the Adam optimizer with default momentum parameters, with a base learning rate of 0.001 that follows the linear scaling rule (L=0.001×Batch_size) as established in Goyal et al.'s work on large minibatch SGD[2]. Regarding the experiments in Table 3, all tests employed the aforementioned learning rate configuration with a fixed batch size of 16.  As described in the manuscript, we maintained consistent network bandwidth across all tests, and applied identical compression ratios (retaining 10% of gradients) for both DGC and DMGC methods.  All other DPGS-related experimental settings remained unchanged from those previously specified.
> For the experiments depicted in Figure 3, DPGS maintained the same configuration while DMGC used a fixed 10% compression ratio, with identical learning rates and optimizer settings as described earlier.  The only modification involved varying the number of nodes, and we recorded training time (not convergence time) at every 10-epoch interval, maintaining a constant batch size of 16 throughout these experiments.
> Concerning Table 4 and Figure 5, the DPGS configuration remained consistent with our baseline setup except for deliberate variations in two parameters: network bandwidth and compression ratio.  These experiments were likewise conducted with a fixed batch size of 16.
>
> We sincerely appreciate your valuable feedback. If you have any further questions about the paper, please do not hesitate to ask us. We would be deeply grateful if you find our revisions satisfactory and might consider adjusting your overall evaluation.
>
> [1]Lin, Y., Han, S., Mao, H., Wang, Y., and Dally, W. J. Deep gradient compression: Reducing the communication bandwidth for distributed training. URL http://arxiv.org/abs/1712.01887.
> [2]Goyal, P., Dollar, P., Girshick, R., Noordhuis, P., Wesolowski, L., Kyrola, A., Tulloch, A., Jia, Y., and He, K. Accurate, large minibatch SGD: Training ImageNet in 1 hour. URL http://arxiv.org/abs/1706.02677.

---

### Decision · Program_Chairs · 2025-05-01

**Decision:**

Accept (poster)

**Comment:**

Existing MIL approaches for Whole Slide Image (WSI) analysis face inefficiency and performance drops due to non-stackable data (varying instance lengths). This paper proposes DPGS, a framework enabling lossless MIL data stacking, paired with DMGC, a joint gradient-model compression method for distributed training. Experiments on Camelyon16 and TCGA-Lung datasets show 31× faster training, 99.2% smaller model communication size, and 9.3% higher accuracy versus baselines.

During rebuttal, the authors effectively addressed reviewer concerns, and the clarifications led to unanimous positive ratings from reviewers. The AC also concurs with the recommendation to accept this paper.